# Pulmonary Diseases in Older Patients: Understanding and Addressing the Challenges

**DOI:** 10.3390/geriatrics9020034

**Published:** 2024-03-07

**Authors:** Pushpa Raj Joshi

**Affiliations:** Institute of General Practice and Family Medicine, Martin-Luther-University Halle-Wittenberg, 06112 Halle (Saale), Germany; pushpa.joshi@medizin.uni-halle.de; Tel.: +49-345-557-5338

**Keywords:** pulmonary disease, public health, aging, diagnosis, management

## Abstract

As the global population ages, pulmonary diseases among older people have emerged as a significant and growing public health concern. The increasing incidence of these conditions has led to higher rates of morbidity and mortality among older adults. This perspective study offers a thorough overview of the prevalent pulmonary diseases affecting the elderly demographic. It delves into the challenges encountered during the diagnosis and management of these conditions in older individuals, considering factors such as comorbidities, functional limitations, and medication complexities. Furthermore, innovative strategies and personalized interventions such as precision medicine, advanced therapies, telemedicine solutions, and patient-centered support systems aimed at enhancing the care provided to older individuals grappling with pulmonary disorders are thoroughly explored. By addressing the unique needs and complexities of this vulnerable population, healthcare systems can strive towards improving outcomes and enhancing the quality of life for elderly individuals affected by pulmonary diseases.

## 1. Introduction

Pulmonary diseases have profound implications for the global aging population. Chronic lower respiratory tract diseases are the third leading cause of mortality among individuals aged 65 years and older [1,2,3,4]. With the global elderly population steadily increasing, the impact of pulmonary diseases on this demographic group is becoming increasingly apparent. According to the data published in the United Nations World Population Prospects, the global population of people aged 65+ years in 2022 was 71 million, accounting for almost 10% of the world’s population [5]. The trend of an increasing elderly population is predicted to continue at the current pace, and by 2050, the 65+ age group is expected to reach the 16% mark globally and will be 24% of the total population by 2100 [5]. Within this context, the prevalence and burden of pulmonary diseases among older people are poised to escalate, necessitating proactive measures to mitigate their impact on public health and healthcare systems. Moreover, as life expectancy continues to rise and populations age, understanding and addressing the complexities of pulmonary diseases in older people become paramount for ensuring overall well-being and quality of life in this demographic.

Given the substantial and rapidly expanding elderly demographic, it is imperative to explore the convoluted changes the body undergoes with age, particularly in relation to the respiratory system [6,7]. Hence, understanding the nuances of the aging process within the lungs is paramount [6]. This understanding forms the basis for delivering individualized and optimal care to meet the distinctive needs of an aging population. By delving into the physiological transformations intrinsic to aging, healthcare providers gain invaluable insights to design precise interventions and personalized treatment regimens. This approach ensures the delivery of the highest standard of care for older patients, addressing their unique health challenges with precision and efficacy.

Addressing this issue, this study emphasizes the intersection of the physiological changes accompanying aging and their direct implications for healthcare delivery. By illuminating these intricacies, the study provides a valuable perspective on the importance of personalized care tailored specifically to the elderly population. This focus on individualized interventions underscores a shift towards more nuanced and effective healthcare practices, representing a significant advancement in geriatric care methodologies.

## 2. Epidemiology and Risk Factors

The epidemiological factors and prevalence rates of pulmonary diseases in the older population vary based on geographic location, socioeconomic factors, and healthcare access [8,9]. However, certain patterns can be identified globally.

Distinguishing the clinical manifestations of pulmonary diseases in the older population involves noting several key differences compared to younger individuals. Older individuals may present with atypical or less pronounced symptoms, with dyspnea often being the predominant symptom, while cough and sputum production might be less prominent [10,11]. Additionally, the presence of multiple comorbidities such as cardiovascular disease, diabetes, and musculoskeletal disorders can complicate the clinical picture, leading to overlapping symptoms and diagnostic challenges [12]. The age-related decline in lung function and muscle strength exacerbates symptoms and impairs the ability to effectively clear respiratory secretions, increasing susceptibility to infections like pneumonia and exacerbations of chronic respiratory conditions such as Chronic Obstructive Pulmonary Disease (COPD) and asthma [6,7,13].

Moreover, older adults may face diagnostic challenges due to age-related changes in chest anatomy and physiology, making the interpretation of imaging studies and pulmonary function tests more challenging, and may be less able to tolerate invasive diagnostic procedures. Age-related changes in pharmacokinetics and pharmacodynamics can also affect the response to medications, with elderly individuals at increased risk of medication interactions and adverse drug reactions [14]. Furthermore, frailty, characterized by reduced physiologic reserve and increased vulnerability to stressors, significantly impacts the clinical course and outcomes of pulmonary diseases in the older population, often resulting in more severe symptoms, slower recovery, and an increased risk of complications [15,16,17]. Understanding these differences in clinical manifestations is crucial for implementing personalized management approaches that address the unique needs and challenges of older patients with pulmonary diseases.

The primary pulmonary diseases, their epidemiology, associated risk factors, and subsequent approaches to management are discussed below.

### 2.1. Chronic Obstructive Pulmonary Disease (COPD)

COPD severely affects the elderly population, profoundly impacting their quality of life, and thereby elevating the levels of morbidity and mortality [16,18]. Studies show that COPD impacts about 10% of individuals aged 40 years and older worldwide [19,20,21]. As people advance in age, the prevalence of COPD surges significantly, with estimates suggesting that approximately 20–25% of individuals over 70 years old may be affected by this condition [22,23]. Despite its widespread prevalence and link with premature aging, COPD often goes undiagnosed. This leads to delayed interventions and management strategies. Diagnosis typically relies on symptoms and spirometry values indicative of airflow obstruction, but the complexity of the condition often complicates its identification [24,25,26]. It is essential to confirm the diagnosis of COPD according to the Global Initiative for Chronic Obstructive Lung Disease (GOLD) guidelines, not only in older individuals with an active or previous smoking habit but also in the broader population of older people. This recommendation is crucial to ensure that they receive appropriate care tailored to their specific needs.

The management of COPD necessitates a multifaceted approach, encompassing both pharmacologic and non-pharmacologic interventions. Numerous studies have validated the effectiveness of such interventions in alleviating COPD symptoms and enhancing the overall well-being of affected individuals [27,28,29,30].

In the realm of COPD management, the appropriate delivery of medications and the mitigation of potential side effects are pivotal considerations. Tailoring drug delivery methods to suit individual patient needs ensures optimal efficacy while minimizing adverse reactions [28,31,32,33,34]. By focusing on these essential aspects, healthcare providers can significantly enhance the lives of COPD patients, mitigating the impact of this challenging condition on the older population.

### 2.2. Asthma

Asthma—a chronic airway disease—has long been associated with younger individuals [35,36,37]. However, emerging evidence from epidemiological studies and clinical observations has shown that asthma affects individuals of all ages, but elderly patients often have a late-onset form, which is underdiagnosed [38,39,40,41,42,43]. For instance, 7.8% of people aged 65+ years in the USA [44] and 5.1% to 8.2% in Europe [45] are reported to have asthma. Moreover, several studies have reported increasing rates of asthma in the elderly population in less developed countries in Africa and Asia [46,47].

In older people, asthma symptoms can be subtle and mimic other age-related conditions such as COPD or heart failure. Chronic cough, shortness of breath, or fatigue might be attributed to aging, leading to underreporting and delayed diagnosis [48,49].

Due to its atypical presentation, comorbidities, and complex interactions, asthma in elderly people demands a comprehensive, patient-centered approach. Hence, healthcare providers must remain vigilant, considering the nuances of aging and its impact on asthma symptoms and treatment. Asthma management in elderly people should be personalized, considering individual health profiles and preferences [50,51]. Patient education is important, empowering the elderly to recognize symptoms, adhere to prescribed medications, and implement action plans, thereby enhancing their quality of life [52,53].

### 2.3. Pneumonia

Pneumonia, an inflammatory lung infection, poses substantial risks for the elderly [54,55,56]. Aging weakens the immune system, making older adults more susceptible to severe pneumonia, often resulting in hospitalization and, in severe cases, mortality [57,58]. According to the Centers for Disease Control and Prevention (CDC), about 150,000 adults aged 65+ years are annually hospitalized due to pneumonia in the USA [59].

Elderly patients with pneumonia often present with atypical symptoms [60,61]. Instead of the classic signs like high fever and productive cough, they may exhibit confusion, lethargy, or worsening of pre-existing conditions [58,60,61,62,63]. These subtle symptoms lead to delayed diagnosis and increased complications. In addition, elderly patients often take multiple medications, leading to polypharmacy [64,65,66]. Therefore, drug interactions and side effects must be carefully managed, ensuring that prescribed antibiotics and supportive medications do not worsen the patient’s overall health.

Elderly patients, especially those in healthcare facilities, are also susceptible to hospital-acquired pneumonia (HAP) [67,68,69]. Preventive strategies, such as proper hygiene, regular turning and mobilization, and early ambulation, are essential in minimizing the risk of HAP among hospitalized elderly individuals [70,71]. Vaccination, especially against common pneumonia-causing bacteria and viruses, is a vital preventive measure [72,73,74]. Healthcare providers must ensure that elderly patients are up to date with their vaccinations, reducing the risk of severe pneumonia episodes. Elderly patients with pneumonia require not only medical expertise but also compassionate care and emotional support [61,75].

However, pneumonia in older adults can vary depending on the underlying conditions, whether respiratory-related or not, of each individual patient. It is impractical to group all older patients with pneumonia together when providing suggestions for care and management due to the diverse range of factors influencing their condition. Engaging family members and caregivers in the care process fosters a supportive environment, positively impacting the patient’s overall recovery.

### 2.4. Interstitial Lung Diseases (ILDs)

Interstitial lung diseases (ILDs) present a complex array of challenges, especially in the elderly population [76,77,78]. These progressive disorders, characterized by the inflammation and scarring of lung tissue, significantly impact respiratory function. Idiopathic pulmonary fibrosis (IPF), a common ILD, primarily affects older adults, especially those over 60 years old [79,80,81]. ILDs often manifest differently in elderly patients. Symptoms might overlap with age-related conditions, leading to misdiagnosis or delayed intervention. Fatigue, breathlessness, and persistent cough, which are common ILD symptoms, can be mistaken for signs of aging, complicating the diagnostic process [82].

Elderly patients with ILDs frequently have multiple comorbidities, such as heart disease or diabetes [83,84]. Managing these conditions alongside ILDs demands a comprehensive and coordinated approach, considering potential interactions between medications and treatment regimens.

Living with a chronic, progressive condition like an ILD can apparently affect the mental health and quality of life of elderly patients [85,86,87]. Psychosocial support, counseling, and involving caregivers are integral components of holistic care, addressing not only the physical but also the emotional well-being of the patient.

### 2.5. Malignant Lung Disease: Lung Cancer

Lung cancer is a pervasive concern among the elderly population, posing significant challenges for both patients and healthcare providers [88,89]. As individuals become older, the risk of developing lung cancer increases, often coupled with underlying health conditions and complex treatment decisions [90]. The majority of lung cancer cases are diagnosed in people over 65 years old due to atypical symptoms and the presence of other age-related health issues. The symptoms are mostly subtle or wrongly attributed to aging, leading to delayed medical attention and diagnosis [88,91]. The median age at the time of diagnosis of lung cancer is 70 years in the USA [88,92,93], as well as in Germany [94].

Elderly patients often have multiple comorbidities such as heart disease or diabetes that complicate treatment decisions [95,96,97,98]. Balancing cancer treatment with the management of these conditions requires a coordinated effort among healthcare providers to minimize risks and optimize outcomes. Moreover, elderly patients are less tolerant to aggressive treatments like surgery, chemotherapy, or radiation due to decreased functional reserves [99,100,101]. Hence, personalized treatment plans, considering the patient’s overall health, preferences and quality of life, are essential. Moreover, involving elderly patients in treatment decisions empowers them and ensures that the chosen approach aligns with their goals and preferences [102]. Open communication, clear information, and involving family members can facilitate informed decision making.

Promoting awareness about lung cancer symptoms and encouraging regular screenings, especially for high-risk individuals such as long-term smokers, can lead to earlier detection [103]. Early-stage diagnosis often allows for more manageable treatment options and improved outcomes [90].

## 3. Frailty and the Characteristics of Older People

When considering the perspective of frailty and the characteristics of older people with pulmonary diseases, several factors come into play, including fragility, functional dependence, comorbidity, pharmacy (medication management), and psychosocial aspects.

### 3.1. Fragility

Older individuals with pulmonary diseases can often experience increased fragility due to the physiological changes associated with aging and the impact of the disease itself [7,104]. Fragility, in this context, refers to a decreased physiological reserve and increased vulnerability to stressors, which can manifest as reduced muscle strength, impaired mobility, and a susceptibility to infections such as pneumonia or exacerbations of COPD [15].

### 3.2. Functional Dependence

Pulmonary diseases, particularly the advanced stages of conditions like COPD or ILD, can lead to functional limitations that result in dependence on others for the activities of daily living [105]. Shortness of breath, fatigue, and decreased exercise tolerance can significantly impact an individual’s ability to perform tasks such as cooking, cleaning, or even personal hygiene [106]. This functional dependence may necessitate caregiver support or modifications to the living environment to ensure the safety and well-being of the individual.

### 3.3. Comorbidity

Older adults with pulmonary diseases often have multiple comorbid conditions, such as cardiovascular disease, diabetes, or osteoporosis, which can further complicate their health status and management [107,108]. These comorbidities may interact with the pulmonary disease, exacerbating symptoms and increasing the risk of adverse outcomes. Managing multiple conditions simultaneously requires a comprehensive and coordinated approach to healthcare, including medication management and lifestyle modifications [109].

### 3.4. Pharmacy

Medication management is crucial for older adults with pulmonary diseases to control symptoms, prevent exacerbations, and improve quality of life [110,111]. However, older individuals are at increased risk of medication-related problems, such as adverse drug reactions, drug interactions, and non-adherence [112,113]. Polypharmacy (the use of multiple medications) is common in this population due to the presence of multiple comorbidities, which further increases the complexity of medication management and the risk of adverse outcomes [114]. Healthcare providers must carefully consider the benefits and risks of each medication and tailor the treatment regimen to the individual’s specific needs and circumstances.

### 3.5. Psychosocial Aspects

Living with a chronic pulmonary disease can have significant psychosocial implications for older adults, including increased levels of anxiety, depression, social isolation, and impaired quality of life [115,116,117]. Breathlessness, fatigue, and limitations in physical functioning can impact social interactions, participation in meaningful activities, and overall well-being [106,118]. Therefore, addressing psychosocial needs is an essential component of comprehensive care for older adults with pulmonary diseases and may involve interventions such as counseling, support groups, pulmonary rehabilitation, and assistance with social services [119,120,121].

## 4. Pathophysiology

A detailed examination of the age-related changes in lung structure and function provides insights into the pathophysiology of pulmonary diseases in older people. Emphasis should be placed on the alterations in lung mechanics, respiratory muscles, and immune responses that predispose older patients to specific respiratory conditions [122,123].

The pathophysiology of pulmonary diseases in older patients is multifaceted, influenced by a combination of age-related changes, weakened immune responses, chronic inflammation, and a lifetime of environmental exposures [6,7,124]. Recognizing these complexities is essential for healthcare providers to develop targeted interventions, personalized treatment plans, and preventive strategies. Table 1 provides a concise summary of the pathophysiology of pulmonary diseases and their impact on patients’ quality of life and overall health. It also outlines various management strategies employed to address these conditions effectively.

## 5. Diagnostic Challenges

The accurate diagnosis of pulmonary diseases in the elderly population can be intricate due to overlapping symptoms, atypical presentations, and limitations in conventional diagnostic tools [138,139]. Diagnosing pulmonary diseases in elderly patients demands a nuanced approach that recognizes the challenges posed by atypical symptoms, comorbidities, and age-related physiological changes [16]. Healthcare providers can navigate these complexities and ensure accurate diagnoses and tailored treatment plans through patient-centered communication, technological advancements, and interdisciplinary collaboration. By effectively addressing these challenges, healthcare professionals can improve the quality of life for elderly patients and enhance their overall respiratory health. Table 2 outlines the common diagnostic challenges encountered in pulmonary diseases and provides subsequent management strategies to address these hurdles effectively.

## 6. Innovations in Treatment and Care

Innovative treatments, such as gene therapy, immunomodulatory agents, and telemedicine, are promising avenues in the management of pulmonary diseases in elderly patients [143,144]. Innovations in the treatment and care of pulmonary diseases are reshaping the landscape of respiratory healthcare for elderly patients. By embracing precision medicine, advanced therapies, telemedicine solutions, and patient-centered support systems, healthcare providers can significantly enhance the quality of life for elderly individuals living with pulmonary conditions [145,146].

Various new treatments have emerged in the field of pulmonary medicine, offering promising prospects for improved management and outcomes of lung diseases. Biological therapies, such as monoclonal antibodies targeting specific inflammatory pathways, have shown potential for reducing exacerbations and enhancing lung function in severe asthma and eosinophilic pulmonary disorders [147,148,149]. Precision medicine approaches, including targeted therapies and genetic-based treatments, are increasingly tailored to individual patients’ genetic profiles, offering more effective and personalized management of conditions like lung cancer and cystic fibrosis [150,151]. Gene therapy techniques aim to correct genetic defects underlying pulmonary diseases such as cystic fibrosis and alpha-1 antitrypsin deficiency by delivering functional genes or gene-editing tools directly to affected cells [152]. Stem cell therapy holds promise for regenerating damaged lung tissue in conditions like COPD and idiopathic pulmonary fibrosis (IPF), with ongoing trials showing potential benefits in improving lung function and quality of life [153,154]. Immunotherapy, traditionally used in cancer treatment, is being explored for certain autoimmune pulmonary disorders to modulate the immune response and reduce inflammation [155]. Advances in inhaler technologies, including smart inhalers and novel drug delivery systems, are enhancing the effectiveness and ease of use of inhaled medications for asthma and COPD [156]. Lung transplantation remains a lifesaving option for end-stage pulmonary diseases, with advancements in surgical techniques and post-transplant care improving outcomes and expanding eligibility criteria [157,158,159].

These innovative treatments signify significant progress in pulmonary medicine, offering new hope for patients with various lung diseases. Continued research and clinical trials are crucial for further refining these treatments and enhancing their effectiveness and accessibility on a global scale. These innovative approaches not only improve clinical outcomes but also promote independence, resilience, and overall well-being, ensuring a fulfilling and comfortable life for elderly patients affected by respiratory diseases (Table 3).

## 7. Management Strategies

Managing pulmonary diseases in elderly patients demands a multifaceted approach that prioritizes individual needs, emphasizes lifestyle modifications, and integrates psychosocial support [166]. By focusing on personalized care plans, regular monitoring, and involving supportive networks, healthcare professionals can significantly enhance the quality of life for elderly patients. Empowering elderly patients to actively participate in their care journey fosters a sense of control and contributes to their overall well-being, ensuring a fulfilling and comfortable life despite the challenges posed by pulmonary diseases [167,168]. Table 4 provides a comprehensive summary of the management strategies for various symptoms associated with pulmonary diseases.

It is essential to note that the prevalence rates of various pulmonary diseases can vary significantly between different countries and regions [9,175,176,177]. Moreover, the accurate assessment of prevalence, diagnosis, and management is often challenging due to underdiagnosis, especially in older populations where symptoms might be attributed to aging rather than specific pulmonary conditions. Regular health check-ups, early diagnosis, and access to proper healthcare facilities are crucial in managing and reducing the impact of pulmonary diseases in the older population [137,178].

## 8. Conclusions

Pulmonary diseases in older patients pose intricate challenges that advocate for a holistic and patient-centered approach in their management. Several studies have underscored the importance of understanding the unique aspects of these conditions within the aging population. By embracing innovative diagnostic tools and treatments, such as precision medicine, advanced therapies, telemedicine solutions, and patient-centered support systems, healthcare professionals can not only provide effective care but also ensure compassion and dignity in the treatment process. The synthesis of existing research emphasizes the urgent need for continued efforts in research, multidisciplinary collaboration, and the implementation of personalized medicine approaches. By prioritizing these strategies, the complexities of pulmonary diseases in the elderly can be navigated, ultimately enhancing their respiratory health and overall well-being.

## Figures and Tables

**Table 1 geriatrics-09-00034-t001:** Pathophysiology of pulmonary disease, their consequences, and subsequent management strategies.

Pathophysiology	Consequences	Management
Reduced lung elasticity [125]	Reduced ability to inhale and exhale	Breathing exercise
Weakened immune response [126]	Susceptible to respiratory infections	Vaccination, vitamin supplement
Altered respiratory muscle function [127]	Diminished strength and endurance of respiratory muscles	Respiratory muscle training [128]
Impaired gas exchange [129]	Decline in the surface area available for oxygen exchange	Exercise to promote optimal lung expansion and ventilation
Inflammaging [130]	Chronic inflammation in the airways	Therapeutic intervention including SIRT1 activators or polyphenols [131]
Comorbidities and multimorbidity [132,133]	Multiple comorbid conditions, such as cardiovascular diseases and diabetes	Care coordination [134,135]
Exposure history [136]	Lifetime of exposure to environmental pollutants, occupational hazards, and cigarette smoke	Lifestyle intervention, awareness [137]

**Table 2 geriatrics-09-00034-t002:** Frequent diagnostic challenges and subsequent management approaches of pulmonary diseases in older patients.

Diagnostic Challenges	Consequences	Solution(s)
Atypical symptoms [140]	Misdiagnosis or delayed diagnosis	Early diagnostic intervention
Comorbidity interference [133]	Heart disease or gastroesophageal reflux diseases exacerbating respiratory symptoms	Careful evaluation and a comprehensive understanding of the patient’s medical history
Age-related physiological changes [6,124]	Reduced lung elasticity and diminished respiratory muscle strength	Understanding the changes to differentiate between normal aging and pathological conditions
Limited exercise capacity [105]	Pulmonary function tests involving exercise might be challenging	Finding suitable alternatives or modifying testing procedures
Technological limitations [141]	Challenges in tolerating various sophisticated tests due to frailty or claustrophobia	Developing patient-friendly protocols and ensuring a supportive environment
Communication barriers [142]	Elderly patients might face difficulties in effectively communicating, especially if they have hearing impairments or cognitive decline	Clear and patient-centered communication strategies, possibly involving family members or caregivers

**Table 3 geriatrics-09-00034-t003:** Contemporary therapeutic interventions in pulmonary diseases.

Interventions	Treatment Plan(s)
Precision medicine [7,160]	Treatment based on genetic, molecular, and lifestyle factors
Biological therapies [161,162]	Specific pathway-targeted monoclonal antibodies and immunomodulatory agents
Inhaled therapies and devices [163]	Innovative smart inhalers with enhanced usability features
Pulmonary rehabilitation programs [164]	Programs combining exercise therapy, nutritional counseling, and psychological support
Robotic-assisted surgery [165]	Modern surgical techniques for precision, minimal invasiveness, and faster recovery
Patient-centered education and support [16]	Interactive educational platforms and mobile applications for regular reminders and lifestyle recommendations

**Table 4 geriatrics-09-00034-t004:** Management approaches for pulmonary diseases.

Approach	Procedure
Comprehensive assessment [169]	Thorough evaluation, considering the specific challenges of older patients
Individualized treatment plans [170]	Tailored treatment plans to a patient’s unique needs
Lifestyle modifications [137]	Smoking cessation programs, pulmonary rehabilitation, regular physical activity, dietary needs
Oxygen therapy [171]	Proper administration and availability of portable oxygen systems
Psychosocial support [167]	Counseling and support groups, addressing anxiety, depression, and coping strategies, involving family members to enhance mental resilience
Regular monitoring [172]	Monitoring lung function, symptoms, and medication adherence, scheduled follow-ups
End-of-life planning [173]	Open discussions about end-of-life preferences and palliative care options
Home healthcare and telemedicine [174]	Continuous monitoring and timely interventions, remote consultations

## Data Availability

No data were used for this perspective.

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
