# Peer review of "Pulmonary Diseases in Older Patients: Understanding and Addressing the Challenges"

_geriatrics, 2024, doi:10.3390/geriatrics9020034_

Round 1
Reviewer 1 Report
Comments and Suggestions for Authors
The manuscript is a narrative review of an important topic in geriatric practice. So it is an article of interest.
But it is not a specific geriatric review, but rather a very general one.
An issue that demonstrates this statement is that it does not even consider frailty in the pathophysiology and in the clinical and management of patients, which totally interferes in the clinical and management aspects of the elderly patient.
Therefore, the article, without this aspect, barely provides information in relation to any geriatrics manual.
We recommend rewriting the total article from the perspective of frailty and the characteristics of older people (fragility, functional dependence, comorbidity, pharmacy and psychosocial aspects) that fully interfere with the management of COPD in older people.
Author Response
Author response to the comments of Reviewer 1
- The manuscript is a narrative review of an important topic in geriatric practice. So it is an article of interest.
Response: Thank you for expressing interest in the submitted article, and I deeply appreciate your constructive feedback aimed at enhancing quality of the manuscript. Your time and effort in reviewing my work and offering valuable insights are truly invaluable to me.
- But it is not a specific geriatric review, but rather a very general one. An issue that demonstrates this statement is that it does not even consider frailty in the pathophysiology and in the clinical and management of patients, which totally interferes in the clinical and management aspects of the elderly patient. Therefore, the article, without this aspect, barely provides information in relation to any geriatrics manual. We recommend rewriting the total article from the perspective of frailty and the characteristics of older people (fragility, functional dependence, comorbidity, pharmacy and psychosocial aspects) that fully interfere with the management of COPD in older people.
Response: Thank you very much for bringing this aspect to my attention. I have now incorporated the discussion on frailty and the characteristics of older individuals into the manuscript as a seperate section (Page: 4-5, lines: 201-252). Additionally, following the suggestions of other reviewers, I have dispersed the relevant information throughout various sections of the manuscript.
Reviewer 2 Report
Comments and Suggestions for Authors
Dear Author,
The topic is interesting, original and in line with the current literature regarding the future recommendations about pulmonary diseases. In my opinion, however, the paper can be accepted after minor revision. The following suggestions can be considered:
Title
It would be better to re-consider the title. The paper could talk about chronic pulmonary diseases and NOT about acute pulmonary disease such as pneumonia.
Epidemiology and risk factors
It would be helpful to divide patients in malignant (lung cancer) and non malignant disease (asthma, COPD, ILD).
2.1 Chronic Obstructive Pulmonary Disease (COPD)
I would mention that the diagnosis of COPD is to be confirmed according to GOLD guidelines also in older people, not only in older individuals with active or previous smoking habit. This recommendation is necessary to provide them appropriate care.
2.3 Pneumonia
Pneumonia in older adults can vary based on the underlying conditions (respiratory or not) of an individual patient. It would be impossible considering all patients in advanced age with pneumonia as a group for providing suggestions of care and management.
Tables
I would suggest that the Tables 1-4 are more attractive.
Comments on the Quality of English Language
Minor editing of English language required.
I would use "older" instead of "elderly".
Author Response
Author response to comments of Reviewer 2
- The topic is interesting, original and in line with the current literature regarding the future recommendations about pulmonary diseases. In my opinion, however, the paper can be accepted after minor revision.
Response: Thank you very much for dedicating your valuable time to reviewing my manuscript. Your positive feedback on my work is greatly appreciated.
The following suggestions can be considered:
- Title: It would be better to re-consider the title. The paper could talk about chronic pulmonary diseases and NOT about acute pulmonary disease such as pneumonia.
Response: The title has been slightly modified. Since the article focuses on pulmonary diseases, omitting pneumonia would create a gap in the message. Therefore, pneumonia is briefly discussed in the manuscript to ensure comprehensive coverage of relevant topics.
- Epidemiology and risk factors: It would be helpful to divide patients in malignant (lung cancer) and non malignant disease (asthma, COPD, ILD).
Response: Since the malignant group consists solely of lung cancer, it is discussed as the Final sub-topic under the category of malignant lung disease (page 4, lines: 177-199).
- 1 Chronic Obstructive Pulmonary Disease (COPD): I would mention that the diagnosis of COPD is to be confirmed according to GOLD guidelines also in older people, not only in older individuals with active or previous smoking habit. This recommendation is necessary to provide them appropriate care.
Response: Thank you fort he suggestion.This message is included in the manuscript (Page:2-3, lines: 95-99).
- 3 Pneumonia: Pneumonia in older adults can vary based on the underlying conditions (respiratory or not) of an individual patient. It would be impossible considering all patients in advanced age with pneumonia as a group for providing suggestions of care and management.
Response: The message is incorporated in the manuscript (Page: 4, lines: 152-155).
- Tables:I would suggest that the Tables 1-4 are more attractive.
Response: The tables have been redesigned to enhance their visual appeal.
- I would use "older" instead of "elderly".
Response: Subsequent changes are done.

Reviewer 3 Report
Comments and Suggestions for Authors
The article covers a very interesting and current topic. Nevertheless, in my opinion, several parts need to be improved, I have some comments:
1) Abstract. Pulmonary diseases among the elderly present a substantial and escalating public health challenge. With the aging of the global population, the incidence of these conditions is surging, resulting in increased morbidity and mortality rates among older adults. This review provides a com-prehensive overview of prevalent pulmonary diseases affecting the elderly population. Further-more, the obstacles encountered during diagnosis and management of the pulmonary diseases in elderly populace are delved, and the innovative strategies and interventions aimed at improving the care provided to elderly individuals dealing with pulmonary disorders are explored. Abstract might be beneficial to include a sentence, that briefly summarizes the key findings of the study. This can provide readers with a quick overview of the research. Abstract might be beneficial to include a sentence that briefly summarizes the key findings of the study. This can provide readers with a quick overview of the research.
2) 1. Introduction 18 Pulmonary diseases have profound implications for the global elderly population. 19 Chronic lower respiratory tract diseases are the third leading cause of mortality among 20 individuals aged 65 years and older [1–4]. With the global elderly population steadily 21 increasing, the impact of pulmonary diseases on this demographic group is becoming 22 increasingly apparent. According to the data published by the United Nations World 23 Population Prospects, the global population of people aged 65+ years in 2022 was 71 mil- 24 lion; accounting for almost 10% of the world’s population [5]. The trend of increasing el- 25 derly population is predicted to continue at the ongoing pace, and by 2050, the 65+ age 26 group is expected to hit the 16% mark globally and will be 24% of the total population in 27 2100 [5]. Please, improve this part and introduce some information regarding the pulmonary diseases in elderly population.
3) This knowledge serves as the foundation for deliver- 32 ing optimal care tailored to the unique needs of the aging population. By comprehend- 33 ing the physiological transformations associated with aging, healthcare providers can 34 develop targeted interventions and personalized treatment strategies, ensuring the high- 35 est standard of care for the elderly patients. Please improve the description of this part and underline the novelty of the study.
4) 2. Epidemiology and risk factors. Please, add some information regarding the differences in clinical manifestations for pulmonary diseases in elderly population.
5) 5. Innovations in treatment and care 179 Innovative treatments, such as gene therapy, immunomodulatory agents, and tele- 180 medicine, are promising avenues in the management of pulmonary diseases in the el- 181 derly patients [119, 120]. I suggest to improve this section and insert some information regarding the different new treatments
6) 7. Conclusion 212 Pulmonary diseases in the elderly patients present intricate challenges necessitating 213 a holistic and patient-centered approach. By understanding the unique aspects of these 214 conditions in the elderly population and embracing innovative diagnostic tools and 215 treatments, healthcare professionals can provide effective and compassionate care, ulti- 216 mately improving the respiratory health and overall well-being of elderly individuals. 217 Geriatrics 2023, 8, x FOR PEER REVIEW 8 of 18 Continued research, multidisciplinary collaboration, and a focus on personalized medi- 218 cine are key to addressing the complexities of pulmonary diseases in the aging popula- 219 tion. The conclusions section needs to be improved. It may be interesting to record the aim of the study. I think it could also be very interesting to discuss the published literature.
Comments on the Quality of English LanguageMinor changes of English language are required
Author Response
Author response to the comments of Reviewer 3
- The article covers a very interesting and current topic.
Response: Thank you very much for reading my manuscript and providing your valuable opinions. I am grateful for your worthy suggestions that has improved the quality of my manuscript.
Nevertheless, in my opinion, several parts need to be improved, I have some comments:
- Pulmonary diseases among the elderly present a substantial and escalating public health challenge. With the aging of the global population, the incidence of these conditions is surging, resulting in increased morbidity and mortality rates among older adults. This review provides a com-prehensive overview of prevalent pulmonary diseases affecting the elderly population. Further-more, the obstacles encountered during diagnosis and management of the pulmonary diseases in elderly populace are delved, and the innovative strategies and interventions aimed at improving the care provided to elderly individuals dealing with pulmonary disorders are explored. Abstract might be beneficial to include a sentence, that briefly summarizes the key findings of the study. This can provide readers with a quick overview of the research. Abstract might be beneficial to include a sentence that briefly summarizes the key findings of the study. This can provide readers with a quick overview of the research.
Response: The Abstract is rewritten summarizing the key findings as suggested by the reviewer.
- Introduction 18 Pulmonary diseases have profound implications for the global elderly population. 19 Chronic lower respiratory tract diseases are the third leading cause of mortality among 20 individuals aged 65 years and older [1–4]. With the global elderly population steadily 21 increasing, the impact of pulmonary diseases on this demographic group is becoming 22 increasingly apparent. According to the data published by the United Nations World 23 Population Prospects, the global population of people aged 65+ years in 2022 was 71 mil- 24 lion; accounting for almost 10% of the world’s population [5]. The trend of increasing el- 25 derly population is predicted to continue at the ongoing pace, and by 2050, the 65+ age 26 group is expected to hit the 16% mark globally and will be 24% of the total population in 27 2100 [5]. Please, improve this part and introduce some information regarding the pulmonary diseases in elderly population.
Response: Thank you for your suggestion. The introduction part is now improved and information regarding the pulmonary disease in older people is included in the introduction section (Page 1, lines: 33-37).
- This knowledge serves as the foundation for deliver- 32 ing optimal care tailored to the unique needs of the aging population. By comprehend- 33 ing the physiological transformations associated with aging, healthcare providers can 34 develop targeted interventions and personalized treatment strategies, ensuring the high- 35 est standard of care for the elderly patients. Please improve the description of this part and underline the novelty of the study.
Response: The description of this part is improved and the novelty oft he study is underlined (Pages 1-2, lines: 42-54).
- Epidemiology and risk factors. Please, add some information regarding the differences in clinical manifestations for pulmonary diseases in elderly population.
Response: Detailed information regarding the differences in clinical manifestations for pulmonary diseases in older people are added in the manuscript (Page 2, Lines 59-81).
- Innovations in treatment and care 179 Innovative treatments, such as gene therapy, immunomodulatory agents, and tele- 180 medicine, are promising avenues in the management of pulmonary diseases in the el- 181 derly patients [119, 120]. I suggest to improve this section and insert some information regarding the different new treatments
Response: The section is improved and the information regarding new innovative treatment intervensions are included (Page:7-8, lines: 294-320).
- Conclusion 212 Pulmonary diseases in the elderly patients present intricate challenges necessitating 213 a holistic and patient-centered approach. By understanding the unique aspects of these 214 conditions in the elderly population and embracing innovative diagnostic tools and 215 treatments, healthcare professionals can provide effective and compassionate care, ulti- 216 mately improving the respiratory health and overall well-being of elderly individuals. 217 Geriatrics 2023, 8, x FOR PEER REVIEW 8 of 18 Continued research, multidisciplinary collaboration, and a focus on personalized medi- 218 cine are key to addressing the complexities of pulmonary diseases in the aging popula- 219 tion. The conclusions section needs to be improved. It may be interesting to record the aim of the study. I think it could also be very interesting to discuss the published literature.
Response: The conclusion section is improved and the aim of the study and the important message oft he study are highlighted.(Page 10, lines: 346-356).

Round 2
Reviewer 1 Report
Comments and Suggestions for Authors
The review has improved with the changes made.
Comments on the Quality of English LanguageThe review has improved with the changes made.
Reviewer 3 Report
Comments and Suggestions for Authors
The manuscript has been improved, as requested. I have no further comments
Comments on the Quality of English LanguageMinor changes of English language are required